# Oxidation and Heat Shock Resistance of Plasma-Sprayed TiC-CoNi Composite Coatings at 900 °C

Jining He [1,*], Baoqiang Li [1], Hongjian Zhao [1,*], Guanya Fu [1], Jiawei Fan [1] and Yanfang Qin [2]

1 Key Lab for New Type of Functional Materials in Hebei Province, School of Materials Science and Engineering, Hebei University of Technology, Tianjin 300132, China; lbq1244962683@163.com (B.L.); fanjiawei@hebut.edu.cn (J.F.)
2 College of Mechanical Engineering, Suzhou University of Science and Technology, Suzhou 215000, China; yfqskd@sina.com
* Correspondence: hejining2016@sina.com (J.H.); zhj_hebut@163.com (H.Z.)

**Abstract:** In this work, the TiC-reinforced CoNi alloy coatings were prepared by the plasma spraying method. Their microstructure, high-temperature oxidation, and thermal shock resistance at 900 °C were studied. The results showed that the CoNi alloy coating exhibited a single phase (c-Co-Ni-Cr-Mo). After adding Ti-graphite mixed powders, the sprayed coating exhibited TiC and $TiO_2$ phases, besides the c-Co-Ni-Cr-Mo matrix phase. For CoNi alloy coating, the main oxidation products were $Cr_2O_3$ and $CoCr_2O_4$ ($NiCr_2O_4$). For TiC-CoNi alloy coating, the main oxidation products were the $TiO_2$ phase, coupled with $Cr_2O_3$ and $CoCr_2O_4$ ($NiCr_2O_4$) phases. The content of oxides increased with the oxidation time. The oxidation weight gain of the TiC-CoNi composite coating was slightly higher than that of the CoNi alloy coating. The formation of TiC could improve the thermal shock resistance of the CoNi alloy coating.

**Keywords:** CoNi alloy coating; TiC; plasma spraying; oxidation; thermal shock resistance





## 1. Introduction

CoNi alloys have been the candidate material for the high-temperature components because of their many excellent properties (such as good corrosion resistance, mechanical performance, good thermal stability, etc.) [1–3]. In general, CoNi alloys are more economical and effective for surface protection as coating materials for high-temperature service parts. At present, many surface coating technologies have been used by researchers to prepare CoNi alloy coatings, such as thermal spraying [4,5], laser cladding [6,7], electrodeposition [8], etc. Thermal spraying is a simple and highly efficient method for preparing surface coatings and is widely used in engineering applications, especially plasma spraying. Yang et al. [9] prepared an $AlSi_{30}Cu_5$ coating on a 6061 aluminum alloy using plasma spraying to provide wear protection. Zhao et al. [10] studied the wear behavior of plasma-sprayed TiCN-Cu coatings depending on the copper content and found that the coating surface crystallization behavior changed, and its fracture toughness also increased, whereas its wear resistance was not improved. Reddy et al. [11] investigated the applications of plasma-sprayed cermet coatings on steel for gas turbines.

Because of the severe working conditions in some industrial applications, the properties of CoNi alloy coatings could not meet the requirement and then restrict their service life. In order to enhance the properties of the CoNi alloy coatings, an effective method is to add the reinforcing phases. The carbides (such as SiC [12], WC [13], TiC [14], etc.), oxides (such as $Al_2O_3$ [15], $Cr_2O_3$ [16], $TiO_2$ [17], etc.), and nitrides (such as TiN [18–20]) were reported in recent years as the reinforcing phases in CoNi alloy coatings. For instance, Zhang et al. [19] prepared TiN-reinforced Ni-Co alloy coatings using the electrodeposition method and found that the coating exhibited a denser microstructure and better performance. Recently, we plasma-sprayed Co/Ni-based alloy coatings and TiC-reinforced Co/Ni-based

composite coatings. The results displayed that the wear resistance of Co/Ni-based alloys was enhanced by in situ TiC phase formation [21,22]. In many engineering applications, high-temperature environments are common, and this requires the surface coatings to have good high-temperature performance, especially high-temperature oxidation behavior and thermal shock resistance. However, few studies on the high-temperature oxidation and thermal shock resistance of the reinforced CoNi alloy coatings have been reported.

Thus, in our present work, the TiC-reinforced CoNi alloy coatings were prepared by the plasma spraying method. Their microstructure, high-temperature oxidation, and thermal shock resistance at 900 °C were studied. The purpose was to evaluate the effect of the TiC-reinforced phase on high-temperature oxidation and the thermal shock resistance of the CoNi alloy coating. The related mechanism was also discussed.

## 2. Experimental Sections

In this experiment, commercial Q235A steels (size: $\Phi$25 mm $\times$ 10 mm) were used as substrates, and their chemical composition is listed in Table 1. The coating preparation was carried out by GP-80 plasma spraying equipment (Jiangsu Yeyuan Spraying Machinery Factory, Wuxi, China) with a spraying distance of 100 mm. The used spraying current was 500 A, and the voltage was 70 V. During the spraying, Ar (flow rate: 80 L/min) and $H_2$ (flow rate: 3 L/min) were used. The feedstock was a mixture of Ti (size: 28 μm), graphite (size: 5 μm), and CoNi alloy (size: 50–100 μm) powders. The preparation process of the powders can be found in Refs. [21,22].

**Table 1.** The chemical composition of Q235A steel.

| Element | C | Mn | Si | S | P | Cr | Ni | Cu | Fe |
|---------|------|------|------|------|------|------|------|------|------|
| % | 0.14–0.22 | 0.3–0.65 | $\leq$0.3 | $\leq$0.05 | $\leq$0.045 | $\leq$0.3 | $\leq$0.3 | $\leq$0.3 | Bal. |

The JEOL Rigaku 2500/PC and Smartlab X-ray diffraction (XRD, Rigaku D/max2500, Rigaku, Tokyo, Japan) analyzers were used for phase analysis of the samples. The measurement angle of 2θ ranged from 20° to 80°, the step size was 0.015°, and the scanning speed was 4°/min. The morphology of the coatings was observed by the S-4800, manufactured by Hitachi, and the Quanta 450 FEG scanning electron microscope (SEM), manufactured by FEI Hong Kong Ltd. (SEM, Quanta 450 FEG scanning electron microscope, FEI, Hong Kong, China).

The coating was oxidized in a box-type resistance furnace. The effect of holding time (1 h, 3 h, 6 h, 12 h, and 24 h) on the oxidation behavior of the coating was investigated at 900 °C. The weight was measured with an electronic balance after coating oxidation. The thermal shock test was carried out as follows: (1) put the sprayed coatings into the resistance furnace (temperature: 900 °C) and hold it for 10 min; (2) take it out and quench it in water; (3) dry its surface with a hair dryer. If there is no abnormality, continue to repeat the test. When the cracks in the coating reach 1/3 of the total area, the coating failure can be determined, and the test is over. The final failure times of the coating were recorded during the test.

## 3. Results and Discussion

The detected phase structure by XRD of plasma-sprayed TiC-CoNi composite coatings is displayed in Figure 1. The coating formed by plasma spraying CoNi alloy powders exhibits a single face-centered cubic phase (c-Co-Ni-Cr-Mo). After adding Ti-graphite mixed powders, the sprayed coating exhibits TiC and $TiO_2$ phases, besides the c-Co-Ni-Cr-Mo matrix phase. The Ti-graphite mixed powders could react to form the TiC phase in the high-temperature plasma flame, and the melted Ti also reacted with oxygen in the air to form the $TiO_2$ phase.

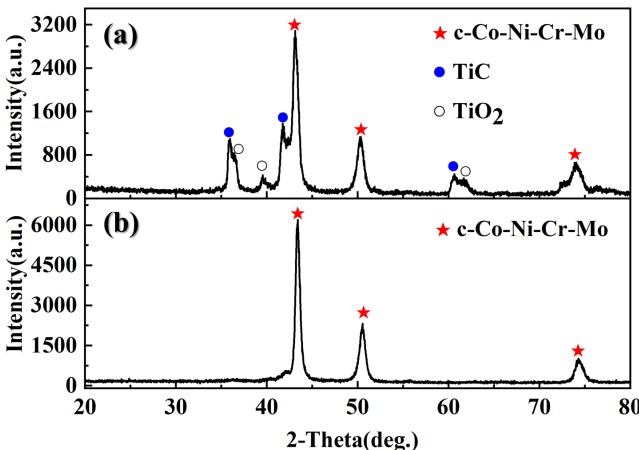

**Figure 1.** The XRD patterns of plasma-sprayed TiC-CoNi composite coatings: (**a**) TiC-CoNi composite coating; (**b**) CoNi alloy coating.

The surface and cross-section micro-morphology observed by SEM of plasma sprayed TiC-CoNi composite coatings are displayed in Figure 2. The pores, particle boundaries, and oxides could be found in the CoNi alloy coating (Figure 2a,(a1),c,(c1)). The whole CoNi alloy coating mainly shows a gray color. For the TiC-CoNi composite coating (Figure 2b,(b1),d,(d1)), it shows gray and dark gray colors (TiC phase). The pores become larger, and the particle boundaries become more obvious. Both the two coatings show a layer structure (Figure 2c,(c1),d,(d1)), which is a typical feature of plasma spraying technology.

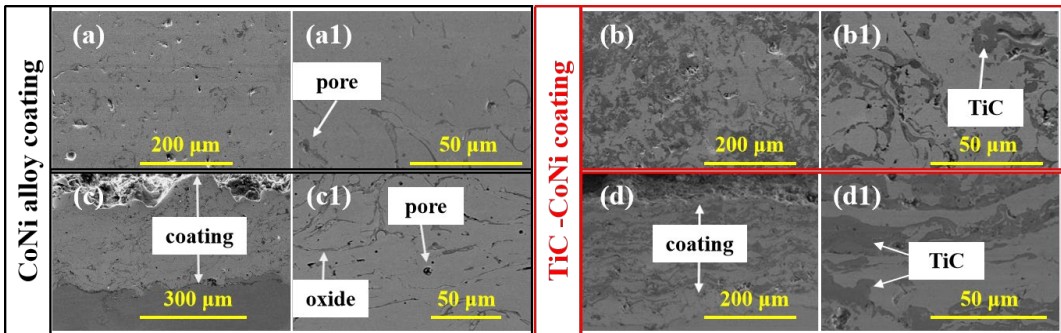

**Figure 2.** The surface/cross-section SEM images of plasma-sprayed TiC-CoNi composite coatings: (**a**) surface of CoNi alloy coating; (**a1**) enlarged image of a; (**b**) surface of TiC-CoNi composite coating; (**b1**) enlarged image of b; (**c**) cross-section of CoNi alloy coating; (**c1**) enlarged image of c; (**d**) cross-section of TiC-CoNi composite coating; and (**d1**) enlarged image of d.

The XRD patterns of TiC-CoNi composite coatings after oxidation at 900 °C are displayed in Figure 3. It can be seen from Figure 3a that the main phase on the surface of the CoNi alloy coating is Co-Ni-Cr-Mo, and the oxidation products are $Cr_2O_3$ and $CoCr_2O_4$ ($NiCr_2O_4$). With the prolongation of the oxidation holding time, the diffraction peak intensities of the oxide phases increase. There is a dynamic change in the relative content of each oxide phase, and the content of $Cr_2O_3$ is slightly higher than that of $CoCr_2O_4$ ($NiCr_2O_4$). From Figure 3b, it can be seen that the main phase of the oxidized TiC-CoNi composite coating is the $TiO_2$ phase, and $Cr_2O_3$ and $CoCr_2O_4$ ($NiCr_2O_4$) phases also formed during oxidation. The diffraction peak intensities of the oxide phases also increase with oxidation time. In addition, the Co-Ni-Cr-Mo phase is still detected in the coating after oxidation for 24 h, indicating that the coatings were not oxidized completely.

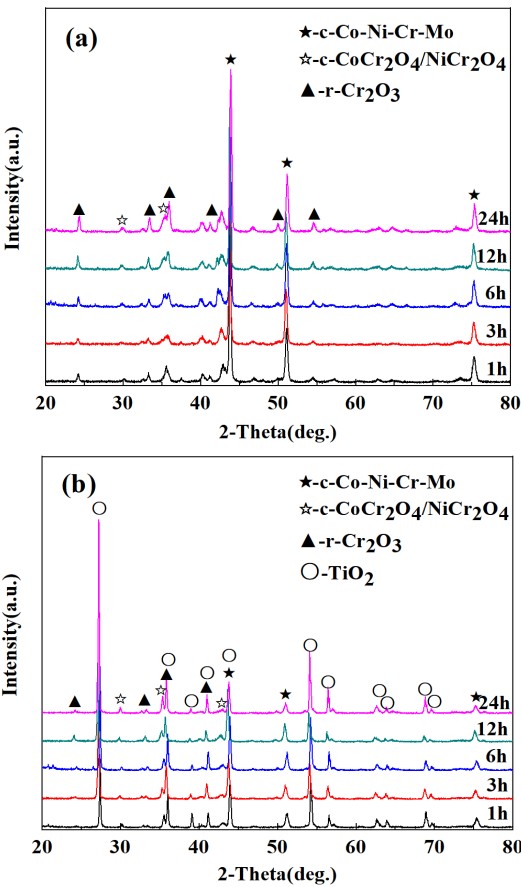

**Figure 3.** The XRD patterns of TiC-CoNi composite coatings after oxidation at 900 °C: (**a**) CoNi alloy coating; (**b**) TiC-CoNi composite coating.

The surface micro-morphologies of TiC-CoNi composite coatings after oxidation at 900 °C for 1 h and 24 h are displayed in Figure 4. It can be seen from Figure 4a,(a1) that the granular oxides appear on the surface of the CoNi alloy coating. The EDS analysis results for A and B areas in Figure 4(a1) are listed in Table 2. Both of these two areas mainly contain O (23.95 at%), Co (27.55 at%), Ni (19.96 at%), Cr (26.30 at%), and Mo (2.24 at%) elements, indicating the coating is oxidizing. Compared with A, the O content (56.06 at%) in the B area is higher, and the Co/Ni content is lower. Based on XRD analysis, the oxides are $Cr_2O_3$ (dominant) and $CoCr_2O_4$ ($NiCr_2O_4$). Compared with 1 h, the size and distribution area of oxides in the coating increased, and some oxides branched with each other (Figure 4(c1)). The EDS analysis results for the A1 area listed in Table 2 show that the granular oxides mainly contain O (60.08 at%), Co (26.46 at%), and Cr (13.46%) elements. For the TiC-CoNi composite coating, after oxidation for 1 h, many dark gray precipitates appear on the coating surface. The EDS analysis results for the C area listed in Table 2 show that the dark gray precipitates mainly contain O (59.46 at%) and Ti (31.08 at%) elements, indicating $TiO_2$ forming. Compared with 1 h, the amounts of dark gray precipitates in the coating increase, and the B1 area also contains mainly O (62.13 at%) and Ti (33.74 at%) elements, indicating the $TiO_2$ phase is increasing as the oxidation time prolongs.

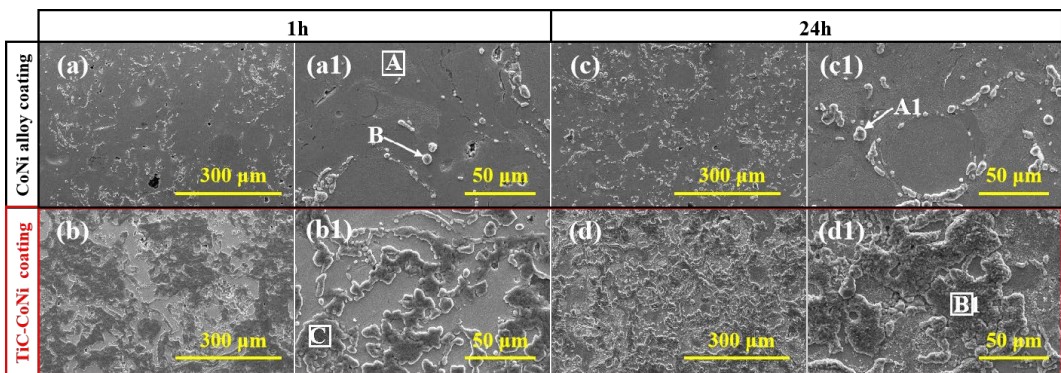

**Figure 4.** The surface micro-morphology of TiC-CoNi composite coatings after oxidation at 900 °C with 1 h ((**a**,**a1**) CoNi; (**b**,**b1**) TiC-CoNi) and 24 h ((**c**,**c1**) CoNi; (**d**,**d1**) TiC-CoNi).

**Table 2.** Energy spectrum analysis results from Figure 4.

| Element | | O | Ti | C | Co | Ni | Cr | Mo |
|---|---|---|---|---|---|---|---|---|
| A | wt% | 7.73 | - | - | 32.74 | 23.63 | 27.58 | 8.32 |
| | at% | 23.95 | - | - | 27.55 | 19.96 | 26.30 | 2.24 |
| B | wt% | 27.03 | - | - | 28.54 | 6.70 | 37.74 | - |
| | at% | 56.06 | - | - | 16.07 | 3.79 | 24.08 | - |
| C | wt% | 36.17 | 56.60 | 3.45 | - | - | 3.79 | - |
| | at% | 59.46 | 31.08 | 7.55 | - | - | 1.92 | - |
| A1 | wt% | 21.73 | - | - | 49.71 | - | 28.56 | - |
| | at% | 60.08 | - | - | 26.46 | - | 13.46 | - |
| B1 | wt% | 36.76 | 59.76 | 1.35 | 2.13 | - | - | - |
| | at% | 62.13 | 33.74 | 3.03 | 1.11 | - | - | - |

Figure 5 shows the time-dependent increase in the coating mass after oxidation at 900 °C. As shown, the quality of the coating increases with the prolongation of the oxidation time. In the early stage of oxidation, the quality of the coating increases rapidly, and then the growth rate gradually flattens. In the early stage of oxidation, because of the high temperature, oxygen could be rapidly adsorbed and ionized on the coating surface, and it would react with the alloy components in the coating to form oxide crystal nuclei. Then, the crystal nuclei grow and form a thin oxide film. The oxidation rate in this stage is controlled by the reaction generated by the alloy element surface, so the oxidation rate is very fast. After the coating surface oxidizes to form a continuous and dense oxide film, the oxidation process enters the second stage. The oxide film on the surface separates the alloy elements in the coating from high-temperature gas-phase oxygen, and the surface oxide film plays a diffusion-blocking role. Further oxidation of the coating requires the outward diffusion of metal cations through the oxide film or the inward diffusion of oxygen ions through the oxide film. The oxidation rate is controlled by the material reaction rate and ion transport rate within the coating/oxide cross section, the oxide film interior, and the oxide/gas phase interface. Therefore, the oxidation rate is significantly reduced, and the growth rate of oxidation weight gain becomes slower. At the same oxidation time, the oxidation weight gain of TiC-CoNi composite coating is slightly higher than that of CoNi alloy coating, which may be due to the higher porosity of the composite coating. In addition, the oxidation of TiC on the coating surface is also the reason for the high oxidation weight gain.

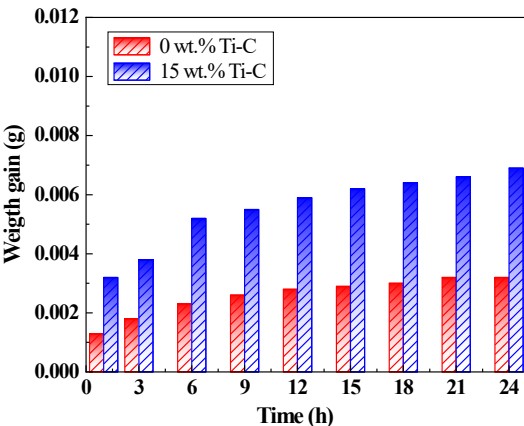

**Figure 5.** The time-dependent increase in the coating mass after oxidation at 900 °C.

When the macroscopic cracks on the coating surface reach one-third of the coating area, this indicates coating failure. The maximum times of thermal shocks before the coating failure could act as an indicator of thermal shock resistance characteristics. In general, the longer the thermal shock times, the better the thermal shock resistance. The thermal shock frequency before coating failure of CoNi composite coatings reaches 34 times, and that of TiC-CoNi composite coatings reaches 47 times. In order to analyze the generation and propagation of cracks during the thermal shock process, the coating surface was observed by SEM, and the results are displayed in Figure 6. Compared with the sprayed state, micro-cracks appear on the surface of the CoNi alloy coating after 20 thermal shocks. As the thermal shock increased to 30 times, the micro-cracks gradually extended, and some micro-cracks were cross-linked with each other. As the thermal shock increases to 50 times, the cracks continue to extend and their width increases, and then the interconnected network cracks form.

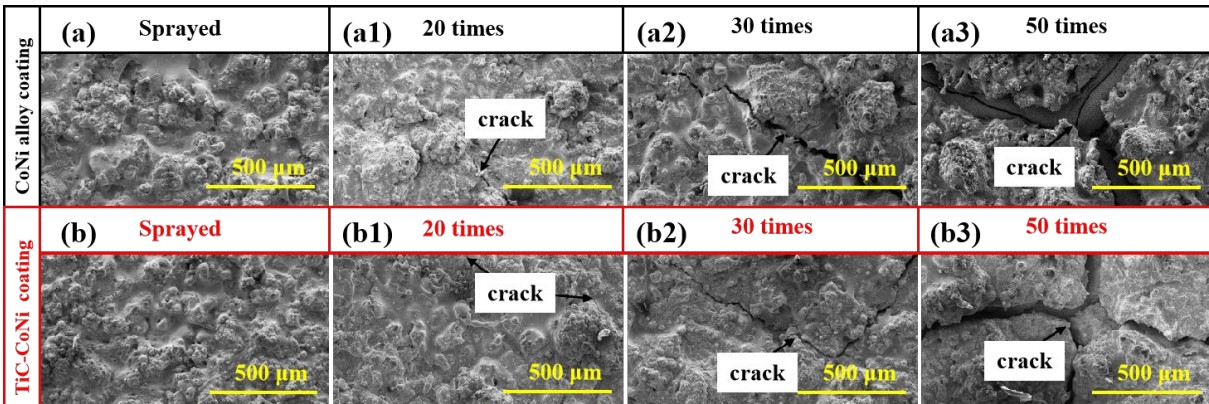

**Figure 6.** The surface morphology of TiC-CoNi composite coatings after thermal shock at 900 °C: (**a**) sprayed CoNi alloy coating; (**a1**) 20 times thermal shock of CoNi; (**a2**) 30 times thermal shock of CoNi; (**a3**) 50 times thermal shock of CoNi; (**b**) sprayed TiC-CoNi coating; (**b1**) 20 times thermal shock of TiC-CoNi; (**b2**) 30 times thermal shock of TiC-CoNi; and (**b3**) 50 times thermal shock of TiC-CoNi.

In general, the thermal shock resistance of coatings is closely related to the cohesion of coatings and the thermal physical properties between coatings and substrates. During the thermal shock test, the main driving forces for crack initiation and propagation in the coating come from the organizational stress caused by material phase transformation and the gradient thermal stress generated by rapid cooling and heating. The result of the superposition of organizational stress and thermal stress is the accumulation of high strain energy within the coating. When the strain energy exceeds the strain tolerance of the coating structure, cracks begin to appear at surface defects and extend along the

particle boundary towards the interior of the substrate. The repeated action of radial stress during the cold-hot impact process could cause the formation of network cracks on the coating surface because of the different thermal expansion coefficients between coatings and substrates. When the temperature rises, the coating surface bears radial compressive stress; when the temperature drops, the surface bears radial tensile stress. Under the repeated impact of rapid cooling and heating, the surface of the coating becomes the most concentrated stress area around the molten particles, and the micro-cracks appear first. With the increase in the number of cycles, cracks continuously initiate and expand, and then the network cracks form. Compared with the CoNi alloy coating, at the same thermal shock times, narrower micro-cracks appear on the surface of the TiC-CoNi alloy coating, indicating better thermal shock resistance. The addition of TiC increases the pores, which could release the stress at the crack tip and then enhance its thermal shock resistance. Moreover, the cohesion of coatings could be improved by forming the TiC phase.

## 4. Conclusions

In our present work, the TiC-reinforced CoNi alloy coatings were prepared by the plasma spraying method. Their microstructure, high-temperature oxidation, and thermal shock resistance at 900 °C were studied. The purpose was to evaluate the effect of the TiC-reinforced phase on high-temperature oxidation and the thermal shock resistance of the CoNi alloy coating. The CoNi alloy coating exhibited a single phase (c-Co-Ni-Cr-Mo). After adding Ti-graphite mixed powders, the coating exhibits TiC and $TiO_2$ phases besides the c-Co-Ni-Cr-Mo matrix phase, indicating TiC and $TiO_2$ phases forming in the high-temperature plasma flame. The main oxidation products are $Cr_2O_3$ and $CoCr_2O_4$ ($NiCr_2O_4$) for CoNi alloy coating, and they are $TiO_2$ (main) and $Cr_2O_3$ and $CoCr_2O_4$ ($NiCr_2O_4$) for TiC-CoNi coating. With the prolongation of the oxidation holding time, the oxide phases increase. The oxidation weight gain of the TiC-CoNi composite coating is slightly higher than that of the CoNi alloy coating. The thermal shock frequency before coating failure of CoNi composite coatings reaches 34 times, and that of TiC-CoNi composite coatings reaches 47 times. The formation of TiC could improve the thermal shock resistance of the CoNi alloy coating but fail to improve its oxidation resistance.

**Author Contributions:** Methodology, H.Z. and G.F.; Software, B.L., H.Z. and Y.Q.; Validation, J.F.; Investigation, B.L., J.F. and Y.Q.; Resources, J.H.; Data curation, G.F.; Writing—original draft, B.L.; Writing—review & editing, H.Z., J.F. and Y.Q.; Supervision, J.H.; Project administration, J.H. and H.Z.; Funding acquisition, J.H. and H.Z. All authors have read and agreed to the published version of the manuscript.

**Funding:** This work was supported by the National Natural Science Foundation of China (No. 51872073).

**Institutional Review Board Statement:** Not applicable.

**Informed Consent Statement:** Not applicable.

**Data Availability Statement:** The data used to support the findings of this study are available from the corresponding author upon request.

**Conflicts of Interest:** The authors declare no conflicts of interest.

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
