# Peer review of "Oxidation and Heat Shock Resistance of Plasma-Sprayed TiC-CoNi Composite Coatings at 900 °C"

_coatings, doi:10.3390/coatings14030296_

Round 1
Reviewer 1 Report
Comments and Suggestions for Authors
I carefully read the entire article, also paid attention to the figures, tables, obtained data and used methods.
During the review of the article, I did not use artificial intelligence or AI-supported tools (such as ChatGPT) to review the submission or create the peer review report.
I assume that my comments are detailed and I hope that the authors will understand and complete or change some points in the manuscript.
The material of the article should be corrected. Check intervals and line spacing, change them if necessary. Figures could be placed in the text. To better access the used literature, it would be necessary to bring their https://doi.org...
In the future, it would be necessary to characterize the materials (besides their coating) at different temperatures, for example 800 C, 900 C, 1000 C, etc. This would give a better insight into the effectiveness of the process, which would need to be shown in the conclusions.
Comments on the Quality of English LanguageHowever, I think that the text of the manuscript should be reviewed by a proofreader who is fluent in English.
Author Response
The material of the article should be corrected. Check intervals and line spacing, change them if necessary. Figures could be placed in the text. To better access the used literature, it would be necessary to bring their https://doi.org... In the future, it would be necessary to characterize the materials (besides their coating) at different temperatures, for example 800 C, 900 C, 1000 C, etc. This would give a better insight into the effectiveness of the process, which would need to be shown in the conclusions. Response: Thank you for your comments. In our study, we have selected the temperature and oxidation time as the influencing factor. In this study, we just investigated the effect of the oxidation time on the CoNi alloy coating and TiC-CoNi alloy coating. As the oxidation temperature increases, the coating oxidation becomes increasingly severe, which is similar to the oxidation time. In addition, we have added the https://doi.org... of the literature.
Reviewer 2 Report
Comments and Suggestions for Authors
Attached File

Neds attantion. Typos exist. Structure is weak in many places.
Author Response
Some typos and language structure need attention. For example “the water” à water and “the oxygen” and similar.
Response: Thank you for your comments. We have checked the whole paper and revised the language structure.
Reconsider the abstract. The sentences from line 9 to line 12 must be in the introduction or delete them.
Response: Thank you for your comments. We have deleted these sentences.
Is it really the c-Co-Ni-Cr-Mo phase. There is not a solid proof for that. XRD shows the peaks but not necessarily what is claimed. Explain the small letter c in the introduction and why this phase is important.
Response: Thank you for your comments. We re-checked the XRD patterns and found that the phase was face-centered cubic Co-Ni-Cr-Mo solid solution, which was represented by c-Co-Ni-Cr-Mo.
Line 60: In this experiment, the Q235 steels, which type of steel did you use A, B, C, or D. Insert the chemical composition of the used steel in a table.
Response: Thank you for your comments. The Q235A steel was used as the substrate materials and we also inserted its chemical composition.
Line 61: Spraying equipment details are needed.
Response: Thank you for your comments. We have added the details of the spraying equipment.
Line 63: Flow rates are very high. Waste of valuables gases. Also safety issues arise. Ar is much heavier than H2.
Response: Thank you for your comments. In our experiment, we used the flow rates to spray coatings because of the following points: (1) the spraying equipment working conditions and the low cost gas; (2) the spraying time is very short (just few minutes).
Line 64: The feedstock was the mixture of Ti (Size: 28 μm), graphite (Size: 5 μm). Give a range for particle size. It is not logical to mention a specific particle size in powder metallurgy. Show SEM images for that and even for CoNi alloy starting powder.
Response: Thank you for your comments. The powder SEM images have been given out in our previous studies (Refs.[21,22]). Here, the particle size was the average values, which we used 500 mesh sieve to screen the powders.
Line 65-66: You need to give more details even though a reference is mentioned. Perhaps any modifications of the preparation process are made for some purpose.
Response: Thank you for your comments. The preparation process are the same and the experiment could be repeated.
Lines 74-77: The procedure is not well described. It is not the style of a scientific paper. It is rather instructions for a technician! Please rephrase all statements and similar in the text.
Response: Thank you for your comments. We have rewritten these sentences.
Lines 77-78: What is the reference for the claimed procedure: ” When the cracks of the coating reach 1/3 of the total area, the coating failure can be determined and the test is ended”?
Response: Thank you for your comments. We have reviewed many references and found that the numerical values defining thermal shock failure of coatings were inconsistent (Applied Surface Science Advances, 2021,3:100043; Open Ceramics, 2023,16:100482;). Here, we claimed this value according to the China standards GB-T 42259-2022.
Line 88: mention a and b in the caption of Fig. 1, what is a and what is b.
Response: Thank you for your comments. We have added the content of a and b.
Lines 89-95: Please rephrase the paragraph. It is not clear. “the” is used without need.
Numbering is not clear. For example, it should read: "a,a1,c,c1” and so on…
Response: Thank you for your comments. We have revised this point.
Line 96: Figure 2 needs to be enlarged, and the scales of the images must stay as original scales of the used SEM.
Response: Thank you for your comments. We have enlarged this figure and the original scales are too small to see clearly.
Line 97: Figure caption should include a,b,c and d description in addition to the
differences/details (surface/ cross-section). Which is which?
Response: Thank you for your comments. We have revised this point.
The same point above applies to the rest of figures. Captions are not detailed! (For example, Fig 3 times need mentioning)
Response: Thank you for your comments. We have revised this point.
Give an example of an image from Figure 4 with its EDS spectrum.
Response: Thank you for your comments. The EDS spectrum was given out in the following figure.
Figure 1. Coating surface energy spectrum after oxidation for 1 h: (a) marked A in Figure 4, (b) marked B in Figure 4, (c) marked C in Figure 4.
In Figure 5, how did the authors determine the weight gain at different times, they did not mention anything about it in the experimental section. You need to justify.
Response: Thank you for your comments. We have added the measure method of the weight of the coatings after oxidation.
Fig. 6 is useless in the presence of Fig. 7. If Fig 6 is needed, then enlarge it and rotate it by 90 degrees to possibly show the color changes as claimed.
Response: Thank you for your comments. We have deleted Fig.6.
Line 205: Is it really: “TiC reinforced CoNi alloy”?
Response: Thank you for your comments. In our work, TiC phase was formed and the micro-hardness of the coating increased. The related data has been published in our previous studies (https://doi.org/10.1016/j.matchemphys.2020.122913). In this view, the TiC reinforced CoNi alloy.
Line 165-166: coating surface changes from light to dark. Why, explain it. Saying that TiC is present is not enough for microstructure analysis.
Response: Thank you for your comments. We have deleted Fig.6 according the above comment and thus the related description also was deleted.
Enlarge Figure 7 and redo the caption to include all details.
Response: Thank you for your comments. We have enlarged Fig.7 (new Fig.6) and redo the captions.
Lines 217-219: “oxidation resistance” is not proven? You need a proper test for that claim.
Response: Thank you for your comments. In order to accurately express the conclusion of the paper, we have modified “oxidation resistance” to “oxidation behavior”.
More points need attention in style and formatting.
Response: Thank you for your comments. We have checked the whole paper and tried our best to improve the quality.

Round 2
Reviewer 2 Report
Comments and Suggestions for Authors
Authors did a good job answering most of the raised points. However, some remarks still need considering:
1. Page 2: line 58: "the chemical composition 58 was listed in Table 1) " it should be is listed. Also, what is the source of the steel.
2. Table 1: Fe % should be stated or mentioned.
3. in Figures: it is better to be consistent where the a and b are. Some a is on top of b and some the opposite.
Comments on the Quality of English Language
Better now.
Author Response
Page 2: line 58: "the chemical composition 58 was listed in Table 1) " it should be is listed. Also, what is the source of the steel.
Response: Thank you for your comments. We have revised this sentence and added the source of the steel.
Table 1: Fe % should be stated or mentioned.
Response: Thank you for your comments. We have added the content (Fe%).
in Figures: it is better to be consistent where the a and b are. Some a is on top of b and some the opposite.
Response: Thank you for your comments. We have tried our best to keep the figures consistent.